# OpenReview forum: "Command-V: Training-Free Representation Finetuning Transfer"
_ICLR.cc/2026/Conference — ICLR 2026 Poster_

### Official Review · Reviewer_inhB · 2025-10-25

**Soundness:** 3
**Presentation:** 3
**Contribution:** 3
**Rating:** 6
**Confidence:** 4

**Summary:**

This paper introduces Command-V, a backpropagation free method which transfers fine tuned behaviors between models by copying a donor model’s residual adapter and pasting its activation level effect into a different recipient model. It builds activation profiles from a prompt set to align layers across architectures and derives simple linear converters that map one model’s hidden space onto another’s. Experiments across model families show that Command-V transfers behaviors such as safety refusal, jailbreak susceptibility, and CoT reasoning.

**Strengths:**

1. Command-V achieves effective behavior transfer without any backpropagation, training data, or gradient updates, making it cheaper and faster than conventional fine tuning.
2. Across diverse tasks, including safety refusal, jailbreak facilitation, and CoT reasoning, the method matches or closely approaches the performance of direct fine tuning.
3. The paper provides a clear mechanism using activation profiling and linear converters, grounding the approach in a mathematically simple yet powerful framework.

**Weaknesses:**

1. The method’s linear converters struggle to bridge major model gaps, e.g., from llama to qwen. So the method strongly depends on strategic model pairing.
2. I think the method might be better for transferring from large model to small model, as in figure 3, qwen and llama from large version to small version have much better performance than from small version to large version. So, it could be difficult to build high-performing generalists by pasting from small specialist, which is a future direction proposed in the discussion.
3. Model D' mentioned on line 138 is not used in the rest parts.

**Questions:**

1. I want to be more precise, does authors only use the representation of the last token of the prompt, without involving any representations from the model’s generated content at all.
2. If we apply this method to transfer different abilities into the same model using multiple converters, would that lead to improvements across all abilities, or could there be conflicts between them? Perhaps it might be worth exploring whether these converters can be merged.

---

> ### Author Response · Authors · 2025-11-21
>
> We thank reviewer inhB for their precise characterization of our work. We address the concerns hereby point by point:
>
> > “The method’s linear converters struggle to bridge major model gaps…”
>
> This is a fair point. However, there are specific instances where we observe successful inter-model-family transfer. For example, Llama-3.2-3B as a recipient model for uncensoring increased from 11% ASR to 58.5% ASR from donor model Qwen2.5-7B. Moreover, we have seen that same-family models that shared pretraining data distribution and other setups result in better ⌘V performance overall. We hypothesize that pretraining-time intervention, potentially with an added objective to match external model activations easily, might bridge this gap and offer a universal donor. This intervention is worthy of its own paper, and on an academic budget, we don’t have the resources to pretrain models of this size, even at the 1B scale, that are similarly performant to models from the Llama and Qwen families.
>
>
> > the method might be better for transferring from large model to small model…  it could be difficult to build high-performing generalists by pasting from small specialist
>
>
> Thanks for pointing this out. We agree in principle, and this is exactly the application that we had envisioned—we believe that the best application of ⌘V is to paste the strong attributes of large models into smaller models that are trained to handle bespoke tasks. This removes the need to apply the same fine-tuning recipe to many smaller models, which may significantly reduce computational costs. This is the same direction as distillation.
> However, we do think that when the loss is acceptable and compute resources (especially GPU memory) are constrained, one can train on the small model and transfer to the large one.
>
> > Model D' mentioned on line 138 is not used in the rest parts.
>
> Thanks for catching the lack of mention for D’. Model D’ only differs from D in terms of the new intervention module ($I$). Everything else is the same. We have clarified the notation in the new draft.
>
>
> >... does authors only use the representation of the last token of the prompt, without involving any representations from the model’s generated content at all
>
>
> Yes, we only use the last token activations. No activations of output tokens are used, and hence any extra latency is one-time per generation.
>
>
> > If we apply this method to transfer different abilities into the same model using multiple converters, would that lead to improvements across all abilities, or could there be conflicts between them?
>
>
> This was actually our original motivation for working on this project! Literature on adapter merging (e.g., Task Arithmetic, Ilharco et al., 2022; TIES-Merging, Yadav et al., 2023) has observed that combining capabilities can be effective, though potential conflicts must be carefully managed. Naively applying them does cause incoherent responses in practice. We are excited to explore ⌘V as an alternative way to merge adapters.
>
> ----
> We would like to thank you again for reviewing our paper and look forward to any further discussions!
>
> References:
> - Ilharco, G., Ribeiro, M. T., Wortsman, M., Gururangan, S., Schmidt, L., Hajishirzi, H., & Farhadi, A. (2023). Editing models with task arithmetic. In Proceedings of the International Conference on Learning Representations (ICLR).
> - Yadav, P., Tam, D., Choshen, L., Raffel, C., & Bansal, M. (2023). TIES-merging: Resolving interference when merging models. In Advances in Neural Information Processing Systems (NeurIPS) (Vol. 36).

---

### Official Review · Reviewer_67Xf · 2025-10-30

**Soundness:** 3
**Presentation:** 3
**Contribution:** 2
**Rating:** 6
**Confidence:** 3

**Summary:**

The paper studies the problem of transferring a representation adapter from a donor LLM to a different recipient LLM without backprop or access to task data. The paper proposes Command-V. Command-V builds a linear converter between matched layers using pseudoinverses. During inference, Command-V applies the donor’s adapter in the recipient’s activation space by first converting the representations to the donor’s space, applying the intervention and then applying them back. Since the converters can be built beforehand, this process process does not require training the recipient model.
The paper evaluates three behaviors, including safety refusal, refusal suppression, and chain-of-thought reasoning. The experimental results show the proposed approach can successfully transfer the representation adapter.

**Strengths:**

The paper studies an interesting problem.

The proposed approach is a straightforward and neat solution for this. And empirically it shows some success.

The experiments are rather thorough (regarding models and tasks), covering multiple behaviors and models.

The paper is well-written and easy to follow.

**Weaknesses:**

While the proposed approach is conceptually interesting, I have some concerns on its utility in practical scenarios:

* There’s some trade-offs between performance and efficiency. In many scenarios the Command-V approach underperforms learning and editing on the recipient itself.
* While the paper argues improved efficiency. It is arguable that this representation editing approach is already very efficient (updating very few parameters, and not requiring tons of data). It is unsure how practitioners will be willing to trade such performance time for efficiency only required at train time.


The experiments only consider one type of representation adaptor. It is unsure how robust the technique is to other editing techniques (like LoFiT (Yin et al., 2025), JoLA (Lai et al., 2025).

**Questions:**

How sensitive is Command-V’s performance to the choice of representation adapter?

---

> ### Author Response · Authors · 2025-11-21
>
> We thank reviewer 67Xf for their review. We address the concerns hereby point by point:
>
> > There’s some trade-offs between performance and efficiency. In many scenarios the Command-V approach underperforms learning and editing on the recipient itself.
>
> > While the paper argues improved efficiency. It is arguable that this representation editing approach is already very efficient … unsure how practitioners will be willing to trade such performance time for efficiency only required at train time.
>
>
> We agree with this! To be clear, our method is not intended as an alternative to training. If resources allow, direct training is naturally superior.
>
> However, training requires access to fine-tuning data and sufficient GPU memory for backpropagation. ⌘V offers significant advantages when these are scarce. It can serve as an efficient alternative to distillation for transferring behaviors from large models to small ones (e.g., for GPU-limited mobile devices), or conversely, allow fine-tuning a small model to transfer behaviors to a larger one.
>
> All of this can be performed entirely on CPU, eliminating the need for forward-backward passes and the associated GPU memory overhead. While ReFT is indeed more efficient than full fine-tuning or LoRA, ⌘V targets a distinct niche: applications where a practitioner may only have access to a limited neural inference processor and CPU. In this setting, ⌘V is the only viable option, making it the most efficient way to edit representations.
>
>
> > “How sensitive is Command-V’s performance to the choice of representation adapter?”
>
> Great question! Hyperparameter-wise, we found that the most important choice of the donor adapter is the number of layers trained. More layers trained usually translate to better performances on the donor model and thus the recipient. Fortunately, varying this hyperparameter does not introduce a significant computation cost.
>
> > … experiments only consider one type of representation adaptor. It is unsure how robust the technique is to other editing techniques (like LoFiT (Yin et al., 2025), JoLA (Lai et al., 2025).
>
> Thanks for pointing us to these references; we have updated our paper to properly cite and discuss these works. Our goal was to give a proof of concept that it is possible to transfer an adaptor between models with very different architectural shapes. We look forward to future work that expands our approach to other types of adaptors. In your referenced adapters, JoLA could benefit from the ⌘V setup by approximating the donor models’ intermediate pre-MLP states, but LoFiT, like LoRA, would require changing the weights of individual attention heads—resulting in a more complicated and computationally costly update.
>
> ----
> Thanks again for engaging with our paper. We would love to answer any additional questions!
>
> References:
> - Lai, W., Fraser, A., & Titov, I. (2025). Joint localization and activation editing for low-resource fine-tuning. In Proceedings of the 42nd International Conference on Machine Learning (ICML).
> - Yin, F., Ye, X., & Durrett, G. (2024). LoFiT: Localized fine-tuning on LLM representations. In Advances in Neural Information Processing Systems (NeurIPS) (Vol. 37).

---

### Official Review · Reviewer_R7Es · 2025-11-04

**Soundness:** 4
**Presentation:** 4
**Contribution:** 4
**Rating:** 10
**Confidence:** 5

**Summary:**

This paper introduces a technique for learning a mapping from one model's representation space to another's that doesn't require training (instead it can be computed analytically with the Moore-Penrose pseudoinverse on a dataset of paired activation samples), and then uses this technique to transfer a learned adapter from one model to another. This approach is compared with learned parameter-efficient finetuning on the target model and seems to perform as well as or even outperform it in some cases on a wide set of tasks.

**Strengths:**

- The method is quite elegant, not requiring any learning but instead computing a transfer matrix analytically with some paired activation data.
- Analyses in Figure 3 and Figure 4 are extremely interesting; it's a novel result in the literature so far that extremely low parameter count PEFTs can learn long-form reasoning, e.g. the ReFT paper showed worse performance on GSM8K in non-reasoning models. It's also interesting how model families don't seem to really capture how well refusal training transfers, particular Qwen models transfer better out-of-family in many cases?
- I really appreciate the analysis of how similar different models' representation spaces are, e.g. Figure 5 in the appendix; these might even be main-text-worthy contributions. There might be applications to understanding model provenance/distillation that go beyond this immediate application to transfer.

**Weaknesses:**

- Some nice connections that might be worth looking at (not really weakness of the paper, just interesting!): There must be some relation to the Platonic Representation Hypothesis ([Huh et al. (2024)](https://arxiv.org/abs/2405.07987)) which will help with framing. Also compare Appendix E.3 of [Wu et al. (2025)](https://arxiv.org/abs/2501.17148) which has some related experiments on affine transfer of steering vectors across models.

**Questions:**

- Why use DiReFT for the adapter as opposed to the other ReFT variants (e.g. LoReFT which outperforms DiReFT)? Is it purely to reduce cost or other performance-related considerations come into play?
- For the ReFT baselines, was LoReFT used?

---

> ### Author Response · Authors · 2025-11-21
>
> We thank reviewer R7Es for their suggestions and questions!
>
>
> > There must be some relation to the Platonic Representation Hypothesis (Huh et al. (2024)) which will help with framing. Also compare Appendix E.3 of Wu et al. (2025) which has some related experiments on affine transfer of steering vectors across models.
>
>
> We are excited to see that recent literature provides strong theoretical and empirical support for our approach!
>
>
> Our findings provide additional evidence for the Platonic Representation Hypothesis (Huh et al., 2024), showing that the shared representation space is robust enough to enable functional transfer of complex behaviors. Huh et al. also introduce concrete convergence metrics (e.g., kernel alignment), and their results suggest these could predict our donor–recipient compatibility ahead of time. This offers a principled way to select strong transfer pairs before running ⌘V, and motivates future work that targets the Platonic space more directly—perhaps by testing the alignment of models to a universal Platonic standard prior to transfer, enabling a generalized behavioral-transplant protocol beyond pairwise mappings.
>
>
> AxBench (Wu et al., 2025) is closely aligned with our setup and suggests complementary directions. Very glad to see this setting is shared! The mentioned Appendix E.3 also shows that steering subspaces can be moved across models via a simple learned affine map (which adds just an extra bias term from our setting).
> The concept-level steering analysis could also motivate a more granular ⌘V: rather than transferring a single adapter, we could decompose it into principal steering directions, transfer only the behaviors we want (e.g., refusal), and drop directions that appear noisy or model-specific.
>
>
> We have updated our draft to include both references.
>
> ----
> > Why use DiReFT for the adapter as opposed to the other ReFT variants (e.g. LoReFT which outperforms DiReFT)? Is it purely to reduce cost or other performance-related considerations come into play?
>
> > For the ReFT baselines, was LoReFT used?
>
>
> We use DiReFT throughout this entire work, as it performs very close to LoReFT and has slightly less latency. Moreover, LoReFT has orthonormal matrices that cannot leverage accelerators in our tested hardware (including Mac M-Series processor), so we opted for the more optimized DiReFT. Finally, while DiReFT is indeed slightly less stable to train than LoReFT, we only applied our method to cases where DiReFT trained successfully.
>
> ----
> Many thanks again for reviewing our paper, and we look forward to any further discussions and suggestions!
>
> References:
> - Huh, M., Cheung, B., Wang, T., & Isola, P. (2024). Position: The Platonic representation hypothesis. In Proceedings of the 41st International Conference on Machine Learning (ICML).
> - Wu, Z., Arora, A., Geiger, A., Wang, Z., Huang, J., Jurafsky, D., Manning, C. D., & Potts, C. (2025). AxBench: Steering LLMs? Even simple baselines outperform sparse autoencoders. In Proceedings of the 42nd International Conference on Machine Learning (ICML).

---

### Official Review · Reviewer_Tbum · 2025-11-07

**Soundness:** 3
**Presentation:** 3
**Contribution:** 3
**Rating:** 6
**Confidence:** 3

**Summary:**

The paper introduces Command-V (⌘V), a training-free approach for transferring fine-tuned behaviors between large language models of varying architectures and sizes.
⌘V aligns the representation spaces of a donor and a recipient model through activation profiles and linear converters, enabling behavioral transfer without access to the original training data and with minimal computational overhead.
Empirical studies across three case scenarios demonstrate that ⌘V achieves performance comparable to direct fine-tuning while consuming significantly fewer computational resources.

**Strengths:**

1. The proposed method avoids the high computational and memory costs typically associated with fine-tuning or distillation.

2. ⌘V requires no access to the task-specific data used to train the donor adapters, making it especially practical in settings where data sharing is restricted or unavailable.

3. Across three distinct case studies, ⌘V has demonstrated competitive performance relative to baseline fine-tuning, with substantially reduced compute requirements.

**Weaknesses:**

1. The approach implicitly assumes a linear correspondence between activation spaces. While computationally efficient, the paper provides limited theoretical or empirical evidence supporting this assumption.

2. ⌘V inherits the strengths and weaknesses of the donor adapter. When the donor yields only modest gains, the transfer conveys little benefit. Moreover, smaller recipient models occasionally experience output collapse or incoherent text generation after transfer.

3. The paper also mentions that predicting good model pairs and task performance transferability remains an open question, posing a challenge for practical deployment.

**Questions:**

1. Could the authors provide stronger theoretical or empirical evidence supporting the validity of the linear converter assumption? In particular, what justifies expecting linear mappings to adequately capture the relationships between donor and recipient activation spaces?

---

> ### Author Response · Authors · 2025-11-21
>
> We thank Reviewer Tbum for probing the validity of our linear correspondence assumption and other feedback. Below we address these concerns point by point.
>
> > The approach implicitly assumes a linear correspondence between activation spaces … limited theoretical or empirical evidence supporting this assumption.
>
> > … what justifies expecting linear mappings to adequately capture the relationships between donor and recipient activation spaces?
>
> Thanks for picking up on this. We certainly agree that the true underlying relationship between the activation spaces of different models is likely nonlinear. We view our approach as approximating this relationship with a linear function—less an assumption than a concrete modeling choice. In this way, a reasonable next step here—and one that we’ve already begun thinking about—is to use non-linear models to paste behaviors. This is a fascinating direction for future work.
>
> As it relates to our response to Review R7Es, our use of linear modeling assumption follows a growing line of concurrent work suggesting that LLMs share aligned representation spaces: Lee et al. (2025) find a “universal” global and local geometry and show that embeddings and unembeddings across model families correspond through simple linear maps. Wolfram & Schein (2025) likewise show that layers at similar relative depths yield aligned activation neighborhoods, implying a conserved representational progression that is well-approximated by linear correspondence.
>
> These results are consistent with broader claims of convergent/“platonic” representations (Huh et al., 2024) and with evidence that representation-level edits and steering directions transfer via linear or affine mappings (Wu et al., 2024; Wu et al., 2025; Bello et al., 2025), providing a concrete empirical basis for the linear converters used in ⌘V. We have added a discussion of these papers to our updated draft in related work.
>
>
> > ⌘V inherits the strengths and weaknesses of the donor adapter. When the donor yields only modest gains, the transfer conveys little benefit.
>
> > Moreover, smaller recipient models occasionally experience output collapse or incoherent text generation after transfer.
>
> We agree that the quality of the donor adapter is critical. ⌘V inherits the donor's characteristics and can suffer from oversteering-like issues, such as output collapse in smaller recipients.
>
> We believe mitigations to oversteering can benefit our work. For example, techniques like classifier-free guidance (Sanchez et al., 2024) have demonstrated effectiveness in avoiding collapsed output. We leave this as future work to consider and have updated our discussion to reflect this.
>
>
> > The paper also mentions that predicting good model pairs and task performance transferability remains an open question, posing a challenge for practical deployment.
>
> We also agree that predicting good model pairs is a key challenge. Our paper shows preliminary indications that compatibility is essential, but this remains an exciting open question.
>
> ----
> We would like to thank you again for reviewing our paper and look forward to any further discussions!
>
> References:
> - Bayat, R., Gholami, A., & Keutzer, K. (2025). Steering large language model activations in sparse spaces. In Proceedings of the Conference on Language Modeling (COLM).
> - Bello, F., Das, A., Zeng, F., Yin, F., & Liu, L. (2025). Linear representation transferability hypothesis: Leveraging small models to steer large models. arXiv preprint arXiv:2506.00653.
> - Huh, M., Cheung, B., Wang, T., & Isola, P. (2024). Position: The Platonic representation hypothesis. In Proceedings of the 41st International Conference on Machine Learning (ICML).
> - Lee, A., Weber, M., Viégas, F., & Wattenberg, M. (2025). Shared global and local geometry of language model embeddings. In Proceedings of the Conference on Language Modeling (COLM).
> - Sanchez, G. V., Hong, S., Jeong, I., & Canny, J. (2024). Stay on topic with classifier-free guidance. In Proceedings of the 41st International Conference on Machine Learning (ICML).
> - Wolfram, C., & Schein, A. (2025). Layers at similar depths generate similar activations across LLM architectures. In Proceedings of the Conference on Language Modeling (COLM).
> - Wu, Z., Arora, A., Geiger, A., Wang, Z., Huang, J., Jurafsky, D., Manning, C. D., & Potts, C. (2025). AxBench: Steering LLMs? Even simple baselines outperform sparse autoencoders. In Proceedings of the 42nd International Conference on Machine Learning (ICML).
> - Wu, Z., Arora, A., Wang, Z., Geiger, A., Jurafsky, D., Manning, C. D., & Potts, C. (2024). ReFT: Representation finetuning for language models. In Advances in Neural Information Processing Systems (NeurIPS) (Vol. 37).

---

### Meta-Review · Area_Chair_VfJF · 2026-01-03

**Summary:**

All reviewers are positive of this paper (even before rebuttal / discussion). They like that the method is cheap, simple/elegant, that it achieves good performance across 3 tasks (with sufficient empirical evidence), and also like some of the additional analyses and figures in the paper. I agree with these points.

Authors rebutted the majority of concerns that the reviewers had. Some of them are natural limitations of the paper that the authors accept and mention as open questions / limitations. Clearly the paper would be better if one or two of those were approached in the paper, but it is still a good paper for ICLR.

**Reviewer Concerns:**

A chief concern that is not addressed is the requirement for good model pairs: which ones might be good and why. Having some initial experiments or hypotheses tested with simple experiments would significantly improve the paper.

Another concern is about using more representation adapters, however, I agree with the authors that this would require more experiments than necessary for a paper, especially given that the authors have a good amount of experiments for the one adapter currently in the paper.

The remaining conerns were addressed sufficiently by the authors (eg linear vs non-linear, can underperform, comparison to training model directly, etc).

**Reviewer Scores:**

I expect that reviewers would keep their scores. Many of the key weaknesses were accepted as limitations of the work, with a decent discussion. These are limitations but not required for acceptance.

---

### Decision · Program_Chairs · 2026-01-26

Accept (Poster)